# Asymmetric Shapley values: incorporating causal knowledge into model-agnostic explainability

**Christopher Frye**

chris.f@faculty.ai

Faculty

**Colin Rowat**

c.rowat@bham.ac.uk

University of Birmingham

**Ilya Feige**

ilya@faculty.ai

Faculty

## Abstract

Explaining AI systems is fundamental both to the development of high performing models and to the trust placed in them by their users. The Shapley framework for explainability has strength in its general applicability combined with its precise, rigorous foundation: it provides a common, model-agnostic language for AI explainability and uniquely satisfies a set of intuitive mathematical axioms. However, Shapley values are too restrictive in one significant regard: they ignore all causal structure in the data. We introduce a less restrictive framework, *Asymmetric Shapley values* (ASVs), which are rigorously founded on a set of axioms, applicable to any AI system, and flexible enough to incorporate any causal structure known to be respected by the data. We demonstrate that ASVs can (i) improve model explanations by incorporating causal information, (ii) provide an unambiguous test for unfair discrimination in model predictions, (iii) enable sequentially incremental explanations in time-series models, and (iv) support feature-selection studies without the need for model retraining.

## 1 Introduction

AI has the capacity to significantly improve economic productivity along with the potential to cause widespread harm to humanity, and the goals of developing AI capabilities while ensuring AI safety are not generally aligned. Helpfully, in the domain of AI explainability, this is not the case. Not only does explainability lie at the heart of AI safety, but it is also critical to the iterative development of new AI systems by exposing how they work, why they fail, and how they can be improved.

Explainability is a nebulous epistemic concept [29] and has been practically approached in many ways. One safe starting point is to restrict one's use to *interpretable* models that require no additional explanation (e.g. linear and rules-based models). This approach is argued for by [34] in particularly sensitive settings. To explain complex models, *model-specific* techniques leverage attributes unique to the model type, e.g. split count for trees [7] or DeepLIFT for networks [36]. However, model-specific approaches are bespoke in nature and do not solve the problem of explainability in general.

*Model-agnostic* methods provide a general approach to explainability that is helpful, not only for its widespread applicability, but also for the common language it provides across model types. E.g. permutation feature importance [6, 39] is a measure that can be meaningfully compared across model types. It serves as a *global* explanation of the model's reliance on each feature in the data. Other methods provide *local* explanations of a prediction on a specific data point [3, 33].

A local model-agnostic approach to explainability based on *Shapley values* is highly compelling due to its principled mathematical foundation [35] and its ability to capture all the interactions between features that lead to a model's prediction. Shapley values have been used in AI explainability for decades [11, 23, 25, 40], with the general framework articulated more recently in [27].

Despite their strengths, Shapley values have 4 outstanding shortcomings: (i) they are computationally expensive, (ii) they rely on unrealistic fictitious data, (iii) they ignore causality, and (iv) they provide explanations based on the raw input features, which may not be amenable to direct interpretation. The computational cost can be reduced, through sampling or with model-specific techniques [26]. A solution to the fictitious data problem is developed in [16]. This paper develops, to our knowledge, the first approach for incorporating causality into the Shapley framework.

Addressing causality in AI explainability should not be considered optional, as causality lies at the heart of understanding any system, AI or otherwise. However, causal deduction is difficult. Indeed, one of the paradigmatic advantages of modern machine learning is its ability to extract highly predictive correlations from large data sets utilising high-capacity models and efficient learning algorithms, sidestepping the necessity for a causal understanding.

The field of *causal inference* does provide a rigorous framework for understanding causality, given a causal graph and some restrictive assumptions [31, 32, 37, 38]. However, the problem of ascertaining the causal graph remains difficult. While *causal discovery* methods exist to automatically extract causal graphs, performance across different methods is highly variable [17]. Moreover, in machine learning, it is exceedingly rare that the full causal model underlying the data is known, since data sets often contain hundreds to thousands of features. Explainability methods should thus incorporate known causal relationships without the prohibitive requirement of a full causal graph. Such a stance is taken e.g. in [28], which aims to generate counterfactual data points that cross class boundaries while preserving a non-exhaustive set of causal constraints.

In this work, we generalise the Shapley-value framework to enable the incorporation of causality. Critically, we do this in a way that preserves the axiomatic construction of the framework, while introducing the opportunity to handle any amount of causal knowledge. In particular, our approach does not require the complete causal graph underlying the data and leads to useful insights even when just a small fraction of causal relationships are known. Our main contributions can be summarised as:

1. By relaxing 1 of 4 axioms underpinning Shapley values, we introduce *Asymmetric Shapley values*: a theoretical framework to integrate causal knowledge into model explainability.

2. We present 4 applications of ASVs: (i) incorporating a partial causal understanding of data into its model's explanation; (ii) a practical test of unfair unresolved bias [20] built into model explanations; (iii) sequential feature importance in time-series modelling; (iv) support for feature-selection through the prediction of model accuracy achievable on a subset of the data's features.

## 2 Shapley values for model explainability

Suppose a team $N = \{1, 2, \ldots, n\}$ of players cooperates to earn value $v(N)$. Here $v$ is a value function [42] associating a real number $v(S)$ with any coalition $S \subseteq N$. *Shapley values* $\phi_v(i)$ offer a well-motivated game-theoretic [35] approach to distributing credit for $v(N)$ among players $i \in N$:

$$\phi_v(i) = \sum_{\pi \in \Pi} \frac{1}{n!} \left[ v(\{j : \pi(j) \leq \pi(i)\}) - v(\{j : \pi(j) < \pi(i)\}) \right] \qquad (1)$$

where $\Pi$ denotes the set of all permutations of $N$, and $\pi(j) < \pi(i)$ means that $j$ precedes $i$ under ordering $\pi$. The Shapley value $\phi_v(i)$ thus represents the marginal contribution that player $i$ makes upon joining the team, averaged over all orderings in which the team can be built.

In the context of supervised learning, let $f_y(x)$ represent a model's predicted probability that data point $x$ belongs to class $y$. If one interprets the input features $\{x_1, \ldots, x_n\}$ as players that cooperate to earn value $f_y(x)$, then Shapley values offer a well-controlled approach to explaining model predictions that appears widely in the machine learning literature [11, 23, 25, 27, 40].

To compute Shapley values for the model prediction $f_y(x)$, one must define a value function $v(S)$ to represent the model's action on a coalition $x_S \subseteq \{x_1, \ldots, x_n\}$ of $x$'s features. It is standard [27] to marginalise unconditionally over out-of-coalition features $x_{\bar{S}} = \{x_1, \ldots, x_n\} \setminus x_S$ as follows:

$$v_{f_y(x)}(S) = \mathbb{E}_{p(x')}\left[ f_y(x_S \sqcup x'_{\bar{S}}) \right] \qquad (2)$$

The expectation is over $p(x')$, the probability distribution from which data is sampled, and $x_S \sqcup x'_{\bar{S}}$ is the spliced data point that combines in-coalition features from $x$ with out-of-coalition features from

$x'$. The value function of Eq. (2) leads directly, through the average over permutations in Eq. (1), to Shapley values $\phi_{f_y(x)}(i)$ that explain the individual prediction $f_y(x)$.

Shapley values stand as the unique attribution method satisfying the following four axioms [35]:

- **Axiom 1 (Efficiency)** $\sum_{i \in N} \phi_v(i) = v(N) - v(\{\})$.
- **Axiom 2 (Linearity)** $\phi_{\alpha u + \beta v} = \alpha\, \phi_u + \beta\, \phi_v$ *for any value functions* $u, v$ *and any* $\alpha, \beta \in \mathbb{R}$.
- **Axiom 3 (Nullity)** $\phi_v(i) = 0$ *whenever* $v(S \cup \{i\}) = v(S)$ *for all* $S \subseteq N \setminus \{i\}$.
- **Axiom 4 (Symmetry)** $\phi_v(i) = \phi_v(j)$ *if* $v(S \cup \{i\}) = v(S \cup \{j\})$ *for all* $S \subseteq N \setminus \{i, j\}$.

In the context of model explainability, Axiom 1 (Efficiency) implies that attribution for the model's output is fully distributed over its input features:

$$\sum_{i \in N} \phi_{f_y(x)}(i) = f_y(x) - \mathbb{E}_{p(x')}\big[f_y(x')\big] \tag{3}$$

with an offset representing the average probability (over all data $x'$) that $f$ assigns to class $y$. This baseline, not attributable to any feature $x_i$, is related to class balance. Axiom 2 (Linearity) means that Shapley values for a linear-ensemble model can be computed as linear combinations of Shapley values for its constituent models. Axiom 3 (Nullity) guarantees that if a feature is completely disconnected from the model's output, it receives zero Shapley value. Axiom 4 (Symmetry) requires attribution to be equally distributed over features that are identically informative of the model's prediction.

The theoretical control offered by these axioms and the consequential uniqueness of Shapley values are coveted properties for many applications. However, we will see in Sec. 3.1 that these four axioms are often too restrictive in the application of model explainability.

## 2.1 On-manifold Shapley values

Shapley explanations are widely based on the value function of Eq. (2). However, the unconditional marginalisation in Eq. (2) is problematic, as out-of-coalition features $x'_{\bar{S}}$ may not be compatible with the in-coalition features $x_S$. For example, in the census data explored in Sec. 4.1, $x_S$ could represent "marital status = never married" and $x'_{\bar{S}}$ could be drawn as "relationship = husband". Such incompatibilities lie *off the data manifold*.

To fix this problem, the value function should involve a conditional marginalisation [19, 41]

$$v_{f_y(x)}(S) = \mathbb{E}_{p(x'|x_S)}\big[f_y(x_S \sqcup x'_{\bar{S}})\big] \tag{4}$$

This *on-manifold* value function is nontrivial to compute, as the empirical approximation to $p(x'|x_S)$ (and distribution-fitting techniques of [1]) cannot be used for high-dimensional data. In [16], two methods are developed to learn the on-manifold value function: a simple-and-direct supervised technique, and a more-flexible unsupervised approach based on variational inference.

The focus of the present work is to incorporate causal knowledge into model-agnostic explainability, and this cannot be done without first getting the correlations right. For this reason, the on-manifold value function of Eq. (4) will be assumed throughout the remainder of the paper.

## 2.2 Global Shapley values

The Shapley values $\phi_{f_y(x)}(i)$ presented above provide a local explanation of the individual prediction $f_y(x)$. *Global Shapley values* [16] for model $f$ are defined by averaging local explanations:

$$\Phi_f(i) = \mathbb{E}_{p(x,y)}\big[\phi_{f_y(x)}(i)\big] \tag{5}$$

over the distribution $p(x, y)$ from which the data is sampled. Global Shapley values explain the model's general behaviour across the data, remaining consistent with the Shapley axioms.

In particular, the global Shapley value $\Phi_f(i)$ can be interpreted as the portion of model $f$'s accuracy attributable to feature $i$. This follows from the sum rule:

$$\sum_{i \in N} \Phi_f(i) = \mathbb{E}_{p(x,y)}\big[f_y(x)\big] - \mathbb{E}_{p(x')}\mathbb{E}_{p(y)}\big[f_y(x')\big] \tag{6}$$

The first term on the right is the accuracy one achieves by sampling labels from $f$'s predicted probability distribution over classes. (Note that this is distinct from the accuracy of predicting the max-probability class.) The offset term is the accuracy one is left with using none of the features: predicting the label of $x$ by sampling from the model's output $f_y(x')$ on randomly drawn $x'$.

# 3 Asymmetric Shapley values

Here we present a theoretical framework to incorporate causal knowledge into model explainability.

## 3.1 Argument against symmetry

In Sec. 2 we discussed the utility of Axioms $1 - 3$ (Efficiency, Linearity, Nullity) satisfied by Shapley values. Axiom 4 (Symmetry) is a reasonable initial expectation: it places all features on equal footing in the model explanation. This forces Shapley values to uniformly distribute feature importance over identically informative (i.e. redundant) features. However, when redundancies exist, we might instead seek a sparser explanation by relaxing Axiom 4.

Consider a model explanation in which Axiom 4 is active, i.e. suppose the value function is symmetric: $v_{f_y(x)}(S \cup i) = v_{f_y(x)}(S \cup j)$ for all $S \subseteq N \setminus \{i, j\}$. Referring to Eq. (4), this means

$$\mathbb{E}_{p(x'|x_{S\cup i})}\big[f_y(x_{S\cup i} \sqcup x'_{\tilde{S}\cup j})\big] = \mathbb{E}_{p(x'|x_{S\cup j})}\big[f_y(x_{S\cup j} \sqcup x'_{\tilde{S}\cup i})\big] \tag{7}$$

where $\tilde{S}$ denotes $N \setminus (S \cup \{i, j\})$. Then, writing the predicted probability $f_y(x)$ as $p(\hat{y} = y|x)$,

$$\int dx'_{\tilde{S}\cup j} \, p(x'_{\tilde{S}\cup j}|x_{S\cup i}) \, p(\hat{y} = y|x_{S\cup i} \sqcup x'_{\tilde{S}\cup j}) \tag{8}$$

$$= \int dx'_{\tilde{S}\cup i} \, p(x'_{\tilde{S}\cup i}|x_{S\cup j}) \, p(\hat{y} = y|x_{S\cup j} \sqcup x'_{\tilde{S}\cup i})$$

which simplifies to

$$p(\hat{y} = y|x_{S\cup i}) = p(\hat{y} = y|x_{S\cup j}) \tag{9}$$

If this holds for all $S \subseteq N \setminus \{i, j\}$ then $x_i$ and $x_j$ contain identical information for predicting model $f$'s output, given any other features $x_S$ that might be known as well.

Eq. (9) could hold trivially if $f_y(x)$ is disconnected from $x_i$ and $x_j$, but this case is covered by Axiom 3 (Nullity). The more common situation is that $x_i$ and $x_j$ are bijectively related to one another. This is exactly the case in which one might want control over which is attributed importance. For example, if $x_i$ is known to be the deterministic causal ancestor of $x_j$, one might want to attribute all the importance to $x_i$ and none to $x_j$, in opposition to Axiom 4 (Symmetry).

## 3.2 Asymmetric Shapley values

We have argued that the requirement of symmetry in model explainability can obfuscate known causal relationships in the data. Interestingly, relaxing this axiom still provides a rich theory. In the game theory literature, this axiom was first relaxed by [30], which termed the result "random-order values"; [43] referred to them as "quasivalues".

Let $\Delta(\Pi)$ be the set of probability measures on $\Pi$, so that each $w \in \Delta(\Pi)$ is a map $w : \Pi \to [0, 1]$ satisfying $\sum_{\pi \in \Pi} w(\pi) = 1$. We define *Asymmetric Shapley values* with respect to $w \in \Delta(\Pi)$:

$$\phi_v^{(w)}(i) = \sum_{\pi \in \Pi} w(\pi) \Big[ v(\{j : \pi(j) \leq \pi(i)\}) - v(\{j : \pi(j) < \pi(i)\}) \Big] \tag{10}$$

ASVs uniquely satisfy Axioms $1 - 3$ (q.v. Theorems 12 and 13 in [30] or Theorem 3 in [43]). They do not satisfy Axiom 4 (Symmetry) unless the distribution $w \in \Delta(\Pi)$ is uniform, in which case they reduce to the Shapley values of Eq. (1).

ASVs thus allow the practitioner to place a non-uniform distribution over the ordering in which features are fed to the model when determining each feature's impact on the model's prediction. While any distribution $w(\pi)$ over orderings provides a model explanation that satisfies Axioms $1 - 3$,

only certain choices of $w(\pi)$ incorporate *causal* understanding into the explanation. To build intuition for the choice of distribution, note that if $w(\pi)$ places nonzero weight only on permutations in which $i$ precedes $j$, then the $i^{\text{th}}$ ASV measures the effect of $x_i$ on the model output assuming $x_j$ is unknown, whereas the $j^{\text{th}}$ ASV measures the effect of $x_j$ assuming $x_i$ is already specified.

This leads to two distinct approaches to causal explainability: one that favours explanations in terms of distal (i.e. root) causes, and one that skews explanations towards proximate (i.e. immediate) causes. The distal approach places weight only on those permutations consistent with known causal orderings:

$$w_{\text{distal}}(\pi) \propto \left\{ \begin{array}{ll} 1 & \text{if } \pi(i) < \pi(j) \text{ for any known} \\ & \text{ancestor } i \text{ of descendant } j \\ 0 & \text{otherwise} \end{array} \right. \tag{11}$$

In this case, the ASVs of known causal ancestors indicate the effect these features have on model output while their descendants remain unspecified; the ASVs of the descendants then represent their incremental effect upon specification. In the proximate approach, $w_{\text{proximate}}(\pi)$ instead places weight on anti-causal orderings, in which $\pi(j) < \pi(i)$ for any known descendant $j$ of ancestor $i$. Note that either case reduces to the uniform weighting $w(\pi) = 1/n!$ of symmetric Shapley values if no causal information is known. We will employ the distal approach in Secs. 4.1 and 4.3 and a variant of the proximate approach in Sec. 4.2.

The distribution $w(\pi)$ thus allows the user to incorporate knowledge of the data's causal structure into explanations of the model's predictions. Note that this is quite distinct from other work [19], which considers the model's prediction process itself to be a causal process (features → model inputs → model output) and finds ordinary Shapley values to be sufficient to explain that process. In contrast, ASVs can incorporate causal structure present in the data itself.

### 3.3 A data agnosticism continuum

Shapley values provide a maximally data-agnostic model explanation by uniformly averaging over all orderings in which features can be introduced. At the other extreme, *causal inference* aims to infer the exact causal process underlying data. ASVs span this data-agnosticism continuum by allowing any knowledge about the data, however incomplete, to be incorporated into an explanation of its model. For example, if causal information is limited, $w_{\text{distal}}(\pi)$ might require that a single known causal ancestor be ordered first, with permutations over remaining features uniformly weighted. Alternatively, if a causal graph is fully specified, $w_{\text{distal}}(\pi)$ might restrict to a single ordering.

ASVs enable some knowledge about the data-generating process to be incorporated into the model explanation, without the often-prohibitive requirement of full causal inference.

## 4 Applications and experimental results

Here we demonstrate that ASVs can offer useful insights: (i) when something is known about the causal structure underlying a model's data, (ii) when there are subtle questions about unfair discrimination sensitive to underlying causality, (iii) when the data type under study possesses intrinsic ordering, and (iv) when one is interested in the predictivity of a subset of the model's features. See App. B for details regarding our implementations, hyperparameters, and uncertainties.

### 4.1 Causality-based model explanations

The ASV framework can lead to useful insights when something is known about the causal structure underlying a model's data. To demonstrate this, we performed experiments using Census Income data from the UCI repository [12]. We trained a neural network to predict whether an individual's income exceeds $50k based on demographic features in the data. Some of the features (e.g. "age") are clear causal ancestors of others (e.g. "education").

First, we computed global (symmetric) Shapley values for this model using the off-manifold value function. This calculation does not respect correlations in the data, evaluating the model on unrealistic splices far from its region of validity. This baseline is labelled "Off manifold" in Fig. 1(a).

Next, we computed global Shapley values that respect correlations in the data, employing the on-manifold value function. This result is labelled "On manifold" in Fig. 1(a).

Finally, we computed global ASVs for this model. We incorporated a basic causal understanding of the data into our choice of distribution $w(\pi)$:

$$w(\pi) = \frac{1}{4! \, 8!} \left\{ \begin{array}{ll} 1 & \text{if } \pi(i) < \pi(j) \text{ for all} \\ & i \in A \text{ and } j \in D \\ 0 & \text{otherwise} \end{array} \right. \qquad \begin{array}{l} A = \{\text{age, sex, native country, race}\} \\ D = \{\text{marital status, education, \ldots, working class}\} \end{array}$$

This follows the distal approach of Eq. (11) as the known causal ancestors $A$ are required to precede all remaining descendants $D$. This result is labelled "ASVs" in Fig. 1(a). ASVs indicate the model accuracy attributable to features in $A$ before the set $D$ is known, as well as the marginal accuracy gained from $D$ assuming $A$ is already known.

Several interesting trends appear in Fig. 1(a). The off-manifold values indicate that the model exhibits strong direct dependence on marital status, more so than on relationship. However, marital status and relationship are so tightly correlated that they give rise to equal on-manifold Shapley values, which correctly take these correlations into account.

ASVs are computed on-manifold as well but place certain features ($A$) ahead of others ($D$) when attributing importance. This results in the ASVs of features in $A$ being greater than or equal to the corresponding on-manifold Shapley values, with strict inequality only when an upstream feature is predictive of a downstream feature that the model depends on directly. One helpful constraint on the explanations of Fig. 1(a) is that they obey Axiom 1 (Efficiency) and thus have equal sums.

Most interestingly, while sex has a relatively small Shapley value, it receives the largest ASV. This means that sex explains enough variation in downstream features – marital status, relationship, occupation, hours per week – to be a strong predictor of income on its own. Quantitatively, the model's accuracy is 85%, and the class balance is 76/24, which prevents us from attributing the majority of this accuracy to any individual feature. ASVs indicate that roughly 3% of the accuracy can be attributed to sex, which is quite significant given the class imbalance. In this way, meaningful insights can be extracted from ASVs with only a crude causal understanding of the data.

## 4.2 Causal explanations of unfair discrimination

Unfair discrimination in machine learning is a pressing concern. Methods exist [14, 15, 18, 44] to impose constraints, e.g. demographic parity, on aggregated model decisions, but many alternative notions of statistical fairness exist [4] and cannot be simultaneously satisfied [10, 22]. More satisfying definitions of fairness that focus on individuals or causality [8, 13, 20, 24] require more machinery to measure or impose, often prohibitively so. See [9] for an introduction to this field. In this section, we demonstrate that ASVs can act as a practical measure of causal unfairness.

Here we focus on *unresolved discrimination* [20], a measure of unfair bias with respect to specific *sensitive attributes*, e.g. gender or ethnicity. This notion does not allow sensitive attributes to influence a model's output unless mediated by *resolving variables*. By definition, a resolving variable may be influenced by sensitive attributes and is itself permitted to influence the model output. For example, in a college admissions process, gender should not directly influence an admission decision, but different genders may apply to departments at different rates, and some departments may be more competitive than others [5]. In this case, department choice acts as a resolving variable: it is influenced by gender, and it influences the admission outcome along a fair channel. If gender were to influence admissions along an unpermitted channel, that would be deemed unfair.

To study unresolved discrimination, we must measure the incremental effect of certain causal ancestors (sensitive attributes) on a model's output, assuming the full effect of specific causal descendants (resolving variables) has already been taken into account. Therefore, a variant of the proximate approach to causal explainability, discussed in Sec. 3.2, is relevant here. In particular, one should take

$$w(\pi) \propto \left\{ \begin{array}{ll} 1 & \text{if } \pi(i) < \pi(j) \text{ for all} \\ & i \in R \text{ and } j \in S \\ 0 & \text{otherwise} \end{array} \right. \tag{12}$$

with $R$ the set of (causally downstream) resolving variables and $S$ the (upstream) sensitive attributes. The resulting ASVs thus measure unresolved discrimination: the influence of sensitive attributes on model output through unpermitted pathways not mediated by resolving variables.

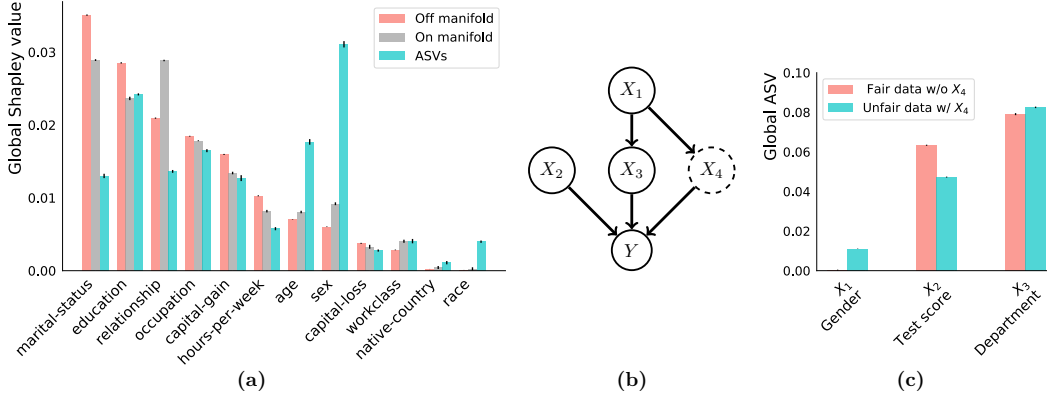

Figure 1: (a) Global SVs and ASVs explaining model trained on Census Income data. (b) Causal graph for generating synthetic college-admissions data sets. (c) ASVs for corresponding models.

To explore this in a controlled setting, we performed experiments on two synthetic college-admissions data sets, which we will refer to as "fair" and "unfair". Each of the data sets contains 3 observed features and a binary label:

$$X_1 = \text{gender}, \quad X_3 = \text{department}, \quad X_2 = \text{test score}, \quad Y = \text{admission},$$

with $X_1$ and $X_3$ binary and $X_2$ continuous. We generated the first data set using the causal graph of Fig. 1(b) but with $X_4$ absent. See App. B.2 for the explicit data generating process. While more men than women are admitted to university in this "fair" data set – 62% versus 38% – this only occurs because a disproportionate fraction of women applied to the more competitive department.

We constructed the second data set using the causal graph of Fig. 1(b). In this graph, there is an additional pathway through which gender can affect admission, through

$$X_4 = \text{unreported referral}. \tag{13}$$

In this "unfair" data set, men at the university recommend other men for admission more often than women. Still, this is not explicit in the data: only $X_1, X_2,$ and $X_3$ are recorded. In this case, 64% of men and 36% of women are admitted, similar to the fair data set. We will show that ASVs have the capacity to verify the fairness of the first data set and expose the unfairness of the second.

To do so, we trained a neural-network classifier on each synthetic data set. Since each classifier varies with all 3 recorded features, symmetric Shapley values are all generically nonzero and incapable of judging whether unfair processes exist in the admissions process.

We computed global ASVs for these models, choosing $w(\pi)$ according to Eq. (12) with $R = \{\text{department}\}$ and $S = \{\text{gender}\}$. The resulting ASVs are shown in Fig. 1(c). The ASV for gender indicates the marginal accuracy gained from gender when department choice is already known, thus indicating whether gender discrimination exists even after department choice is accounted for. This framework verifies the fairness of the data generated without $X_4$, by assigning a vanishing ASV to gender, while exposing the unfairness of Fig. 1(b).

### 4.3 Data types with intrinsic ordering

Next we demonstrate that ASVs offer a natural framework for explainability when data is intrinsically ordered, as they address the sequential incrementality of the model's prediction. To do so, we use Epileptic Seizure Recognition data [2, 12] containing EEG signals and binary labels indicating seizure activity; see e.g. Fig. 2(a). Each time series in the data represents 1 second of an EEG signal, whereas most seizures last 30–120 seconds, so a seizure is occurring (or not) for the entirety of each time series. We trained an RNN to classify time series according to seizure activity.

As a baseline explanation, we first computed global (symmetric) Shapley values for this model, using the off-manifold value function. The result is labelled "SVs" in Fig. 2(b).

The ASV framework can incorporate the natural ordering of the time series to obtain a sparser explanation. We computed global ASVs for this model, choosing $w(\pi)$ according to the distal

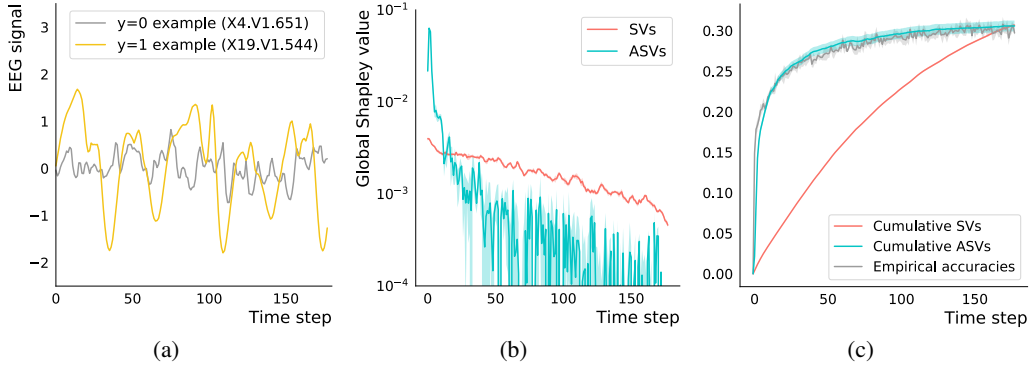

Figure 2: (a) Example EEGs: gray benign, yellow malignant. (b) Global explanations of a model trained on EEG signals. (c) Cumulative sum of the individual values in Fig. 2(b) compared to the empirical accuracy of a model trained on features $t' \leq t$.

approach of Sec. 3.2, i.e. to enforce time-ordering:

$$w(\pi) = \begin{cases} 1 & \text{if } \pi(t_i) < \pi(t_j) \text{ for all } t_i < t_j \\ 0 & \text{otherwise} \end{cases} \tag{14}$$

This result is labelled "ASVs" in Fig. 2(b). These ASVs indicate the additional predictivity gained by time step $t$ assuming steps $t' < t$ are already known.

Note that ASVs concentrate feature importance into the beginning of the time series, while Shapley values attribute significant importance across the entire signal. Indeed, ASVs drop by a factor of roughly $10^2$ after 25 (out of 178) steps, while Shapley values do not change by a full order of magnitude over the entire series. This means that steps 26 through 178 offer little additional predictive power once 1 through 25 are known. Regardless, Axiom 4 (Symmetry) does not allow Shapley values to play favourites: each EEG signal is an arbitrary snapshot of continuous brain activity, so no individual region of the signal is systematically more predictive than another. Shapley values thus spread the model explanation out over all predictive features. ASVs, by contrast, offer an explanation that is more sparse, and more fundamental to the sequential nature of the data.

### 4.4 Precise, verifiable feature selection

Finally, we demonstrate that ASVs support a direct interpretation as the accuracy achievable by a model that uses only a subset of the data's features. This makes ASVs applicable to feature-selection studies that aim to eliminate non-predictive features from a data set.

To set this up, suppose we define ASVs for a model $f$ by choosing

$$w(\pi) = \frac{1}{|U|! \, |V|!} \cdot \begin{cases} 1 & \text{if } \pi(i) < \pi(j) \text{ for all } i \in U \text{ and } j \in V \\ 0 & \text{otherwise} \end{cases} \tag{15}$$

for some partition $U \sqcup V = N$ of the data's features. Then the sum of global ASVs over $i \in U$ is

$$\sum_{i \in U} \Phi_f^{(w)}(i) = A_f(U) - A_f(\{\}) \tag{16}$$

where $A_f(S)$ is the accuracy (in the sense of Sec. 2.2) achieved by $f$ upon marginalisation over features absent from $S$. This sum over $i \in U$ is thus the accuracy that model $f$ gains using features in $U$ but not $V$. The sum of global ASVs over $i \in V$ gives the remainder, i.e. the marginal improvement in accuracy that $f$ achieves using features in $V$ in addition to $U$:

$$\sum_{i \in V} \Phi_f^{(w)}(i) = A_f(U \sqcup V) - A_f(U) \tag{17}$$

In the non-parametric limit, the accuracy of $f$ after marginalisation over $V$ is equal to the accuracy achievable by another model trained solely on $U$. Thus, for a feature-selection study, Eq. (17) represents the decrease in accuracy one could expect upon dropping the features in $V$.

Results analogous to Eqs. (16) and (17) hold if we instead partition the features into many disjoint subsets $U_1 \sqcup U_2 \sqcup U_3 \sqcup \cdots = N$. We can demonstrate this using the model, data, and global explanations of Sec. 4.3. In that case, the ASV of feature $t$ corresponds to the marginal increase in accuracy achieved by the model by accepting feature $t$ on top of features $t' < t$:

$$\Phi_f^{(w)}(t) = A_f(\{t' \leq t\}) - A_f(\{t' < t\}) \tag{18}$$

Cumulative ASVs thus telescope and obey:

$$\sum_{t' \leq t} \Phi_f^{(w)}(t') = A_f(\{t' \leq t\}) - A_f(\{\}) \tag{19}$$

To test this assertion, we trained many models on the EEG data: a separate classifier $f^{(t)}$ for each time step $t$, trained on all previous steps $t' \leq t$. Fig. 2(c) displays the accuracy, in the sense of Eq. (19), of each of these models in gray. These empirical accuracies are plotted alongside cumulative Shapley values and cumulative ASVs, i.e. sums of the individual values from Fig. 2(b).

The close relationship between empirical accuracies and ASVs in Fig. 2(c) demonstrates that ASVs have a precise interpretation as the model accuracy attributable to each feature in the data. This makes them useful for feature-selection studies as they allow the practitioner to avoid re-training many models $f^{(t)}$ on subsets of the data's features.

## 5   Conclusion

In this work, we introduced Asymmetric Shapley values, a mathematically principled framework for model-agnostic explainability that generalises Shapley values to incorporate causal information underlying the model's data. We showed how this framework can be employed across multiple applications: incorporating causal dependencies into model explanations, testing for fairness amidst subtleties like resolving variables, constructing sequential explanations in the context of time series, and selecting important features without model retraining. We hope that lowering the barrier to incorporating causality in AI explainability will lead to the development of better models and the deployment of more trustworthy AI systems throughout society.

### Broader impact

Asymmetric Shapley values provide a method for incorporating causal knowledge into model-agnostic explainability. Like any model-agnostic method, ASVs can be applied to a wide variety of machine learning models, thus creating the potential for broad impact. Increased transparency in algorithmic decision-making enables practitioners to avoid failure modes and build safer models. ASVs in particular allow users to investigate nuanced causal notions of unfairness in models (as in Sec. 4.2) that other explainability methods cannot detect. In this sense, one of the primary applications of ASVs is aimed at preventing malignant societal effects of automated decisions.

Progress in explainability could also conceivably lead to a set of negative outcomes, broadly resulting from blind trust being placed in model explanations. To avoid this, regulatory bodies should not approve consequential decision-making algorithms just because a model explanation has been provided. Furthermore, model explainability should not be considered a replacement for the domain expertise of model developers.

For ASVs in particular, users should be careful only to incorporate causal information after verification by a domain expert, as the use of incorrect causal relationships would negate the benefits of our approach to explainability. We do not view this as a flaw of the framework, but instead as inherent to its flexibility. ASVs grant practitioners the freedom to incorporate any amount of causal information; this necessarily entails a responsibility to do so correctly.

### Acknowledgments and Disclosure of Funding

This work was developed and experiments were run on the Faculty Platform for machine learning. The authors benefited from discussions with Christiane Ahlheim, Tom Begley, Julian Berman, Laurence Cowton, Markus Kunesch, Omar Sosa Rodriguez, Ron Smith, Megan Stanley, and Naoki Yoshihara. C.R. is grateful to Birkbeck College for its hospitality. The authors received no funding to disclose.

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
