[Supplementary Material]



Figure 3: Causal graphs depicting 3 data generating processes.

# A    Examples in two dimensions

Let $n = 2$ and consider a model $f(x_1, x_2)$ trained to approximate a data distribution $p(Y = y | X_1 = x_1, X_2 = x_2)$. The value function of Eq. (4) in this case becomes

$$v(\{\}) = \mathbb{E}_{p(x'_1, x'_2)}[f_y(x'_1, x'_2)] \qquad v(\{1\}) = \mathbb{E}_{p(x'_2 | x_1)}[f_y(x_1, x'_2)] \qquad (20)$$
$$v(\{1, 2\}) = f_y(x_1, x_2) \qquad v(\{2\}) = \mathbb{E}_{p(x'_1 | x_2)}[f_y(x'_1, x_2)]$$

Local Shapley values then follow from Eq. (1):

$$\phi_v(1) = \frac{1}{2}\big[v(\{1\}) - v(\{\})\big] + \frac{1}{2}\big[v(\{1, 2\}) - v(\{2\})\big]$$

$$\phi_v(2) = \frac{1}{2}\big[v(\{1, 2\}) - v(\{1\})\big] + \frac{1}{2}\big[v(\{2\}) - v(\{\})\big]$$

The first bracketed term in each equation corresponds to the permutation $\pi = (12)$, the second to $\pi = (21)$. Axiom 4 (Symmetry) requires these to be weighted evenly.

ASVs provide an additional layer of flexibility, allowing the practitioner to choose $w(\pi)$ according to the application:

$$\phi_v^{(w)}(1) = w_{(12)}\big[v(\{1\}) - v(\{\})\big] + w_{(21)}\big[v(\{1, 2\}) - v(\{2\})\big] \qquad (21)$$
$$\phi_v^{(w)}(2) = w_{(12)}\big[v(\{1, 2\}) - v(\{1\})\big] + w_{(21)}\big[v(\{2\}) - v(\{\})\big]$$

where $w_{(12)} + w_{(21)} = 1$ to satisfy Axiom 1.

Now suppose some basic causal information underlying the data is known. If the data is generated according to Fig. 3(a) then one might choose $w_{(12)} = 1$ and $w_{(21)} = 0$:

$$\phi_v^{(w)}(1) = v(\{1\}) - v(\{\}) \qquad \phi_v^{(w)}(2) = v(\{1, 2\}) - v(\{1\}) \qquad (22)$$

$\phi_v^{(w)}(1)$ thus reports the impact $x_1$ has on model $f$ over its average output, while $\phi_v^{(w)}(2)$ reports the marginal effect of $x_2$ on $f$ after receiving $x_1$. If instead the data is generated by Fig. 3(b) one might choose $w_{(12)} = w_{(21)} = 1/2$. Following Eq. (11), one would set $w_{(12)} = 1$ and $w_{(21)} = 0$ in the case of Fig. 3(c) as well.

# B    Details of experiments

## B.1    Experiment on Census Income data

For the experiment of Sec. 4.1, we used the Census Income data from the UCI repository [12], ignoring the "fnlwgt" feature. The model-to-explain $f$ was a dense network, with 2 hidden layers of 100 units. Using a 75/25 train/test split, the model was trained with default sklearn settings, using early stopping on a validation fraction of 25%. While the data has a 76/24 class balance, the model $f$ achieves 84.7% test-set accuracy. The results in Sec. 4.1 were computed on the test set.

Three variants of global Shapley values appear in Fig. 1(a). Each is an aggregation of local values defined according to Eq. (5). The first variant, labelled "Off manifold", is the standard one, defined

with the off-manifold value function of Eq. (2). We obtained a Monte Carlo estimate of this quantity with $10^6$ samples, plotting the resulting mean as the bar length in Fig. 1(a) and the standard error of the mean as the error bar.

The "On manifold" and "ASV" results in Fig. 1(a) are similarly Monte Carlo estimates with $10^6$ samples. These quantities are defined with respect to the on-manifold value function of Eq. (4). We computed the conditional distribution $p(x'|x_S)$ that appears in this value function using the VAE-based method of [16]. In particular, we used dense neural networks for the encoder, decoder, and masked encoder, each with 2 hidden layers of 100 units, trained using Adam [21] for optimisation, a batch size of 128, and early stopping with a validation fraction of 25% and patience of 20 epochs. No hyperparameter tuning was performed.

## B.2 Experiment on synthetic college admissions data

For the experiment of Sec. 4.2, we used two synthetic college-admissions data sets, which we refer to as "fair" and "unfair", with data generating processes described qualitatively in Fig. 1(b). In both data sets, gender $X_1$ is a binary random variable, with $X_1 = 0$ for women and $X_1 = 1$ for men. It is drawn according to

$$P(X_1 = 1) = 1/2 \tag{23}$$

Test score, $X_2$, is a normally distributed random variable:

$$x_2 \sim N(0, 1) \tag{24}$$

Department choice, $X_3$, is a binary variable drawn differently for women and men:

$$P(X_3 = 1|X_1 = 0) = 0.8 \tag{25}$$
$$P(X_3 = 1|X_1 = 1) = 0.2$$

so that women mostly apply to department $X_3 = 1$ and men to $X_3 = 0$. College admission, $Y$, is a binary variable drawn differently for the two data sets. In the fair case,

$$P(Y = 1|X_2 = x_2, X_3 = x_3) = \text{sigmoid}(x_2 + 2\,x_3 - 1) \tag{26}$$

making $X_3 = 1$ the more competitive department. In the unfair case, admission is additionally based on (binary) unreported referrals, $X_4$, which are more prevalent for men than for women:

$$P(X_4 = 1|X_1 = 0) = 1/3 \tag{27}$$
$$P(X_4 = 1|X_1 = 1) = 2/3$$

While $X_4$ is not reported in the unfair data set, it has an important effect on admissions:

$$P(Y = 1|X_2 = x_2, X_3 = x_3, X_4 = x_4) = \text{sigmoid}(x_2 + 2\,x_3 + 2\,x_4 - 2) \tag{28}$$

Models-to-explain $f^{(\text{fair})}$ and $f^{(\text{unfair})}$ were fit to the two synthetic data sets. For each we used a densely connected network with 2 hidden layers of 10 units. Using a 75/25 train/test split, each model was trained with default sklearn settings, using early stopping on a validation fraction of 25%. While each data set has a 50/50 class balance, $f^{(\text{fair})}$ and $f^{(\text{unfair})}$ achieve 73.6% and 73.2% test-set accuracy, respectively. All results in Sec. 4.2 were computed on held-out test sets.

As in App. B.1, we used the VAE-based method of [16] to compute global ASVs for Fig. 1(c). We used dense neural networks for the encoder, decoder, and masked encoder, each with 2 hidden layers of 20 units, trained using Adam [21] for optimisation, a batch size of 128, and early stopping with a validation fraction of 25% and patience of 20 epochs. No hyperparameter tuning was performed. Bar lengths in Fig. 1(c) correspond to means, and error bars to standard errors, with $10^6$ Monte Carlo samples.

## B.3 Experiments on Seizure Recognition data

The experiments of Secs. 4.3 and 4.4 were performed on the Epileptic Seizure Recognition data [2] from the UCI repository [12]. The model-to-explain $f$ was a recurrent neural network: an LSTM with 20-dimensional hidden state. Using a 75/25 train/test split, the model was trained using Adam [21] for optimisation, a batch size of 128, and early stopping with a validation fraction of 25% and

patience of 20 epochs. While the data has an 80/20 class balance, the model $f$ achieved 98.3% test-set accuracy. All results shown in Sec. 4.3 were computed on the test set.

The global Shapley values in Fig. 2(b) were computed using $10^5$ Monte Carlo samples from the off-manifold value function of Eq. (2). Central values in Figs. 2(b) and 2(c) represent means, with shaded uncertainty bands for the standard error of the means. The global ASVs in Fig. 2(b) were computed similarly, but with the VAE-based method of [16] used to compute the on-manifold value function. We used LSTMs for the encoder, decoder, and masked encoder, each with a 20-dimensional hidden state, trained using Adam [21] for optimisation, a batch size of 512, and early stopping with a validation fraction of 25% and patience of 100 epochs. We varied hyperparameters (for LSTM dimension, batch size, and patience) up and down by a factor of 2 without significant effect.

As described in Sec. 4.4, Fig. 2(c) displays the cumulative values corresponding to Fig. 2(b). Fig. 2(c) also displays a gray curve labelled "Empirical accuracies". For each point $t$ on this curve, a model $f^{(t)}$ was trained to use the restricted time steps $t' \leq t$ to perform the binary classification task. These models were defined and trained identically to the model-to-explain $f$ described above. We computed $A_{f^{(t)}}(\{t' \leq t\}) - A_{f^{(t)}}(\{\})$ for each model, with $A_{f^{(t)}}(S)$ defined in Sec. 4.4. These differences are labelled "Empirical accuracies" in Fig. 2(c) and can be interpreted as the accuracies, above a randomised baseline, achieved by models $f^{(t)}$. We performed 5 trials for each value of $t$. Means appear as the central curve and standard deviations as the shaded uncertainty band.