[Reviews · NeurIPS 2020]

Review 1

Summary and Contributions: The authors present a variation on Shapley values, Asymmetric Shapley values (ASV), that can be used to integrate causal knowledge into model explanations. The authors restricting the set of permutations that are used to define a feature's average marginal contribution in order to to place more emphasis on causal ancestors, which can lead to sparser explanations, and explanations that put appropriate weight on sensitive attributes (like gender or race).

Strengths: - The authors suggest using ASVs, which relax the symmetry axiom, in order to place more importance on causal ancestors in local model explanations. - Their framework of excluding permutations that violate the feature ordering permits a flexible integration of causal knowledge. Causal relationships can be represented by a precise directed graphical model, or by an ordered partitioning. ASVs are therefore applicable even when the precise causal graph is not known. - The experiments demonstrate a couple interesting applications of ASVs, and how they can be used to inflate the amount of influence given to sensitive features.

Weaknesses: About the authors' contribution in introducing ASVs: - It seems like ASVs are giving a new name to a concept that's been studied in game theory for decades: quasivalues (or equivalently, random-order values). Quasivalues are based on precisely the same idea of relaxing the symmetry axiom, and they use very similar notation (see e.g., [1]) to denote the exact same thing as ASVs. - Bringing these into model explanation is a good idea, but section 3.2 is a bit unclear about the origins. The authors do cite relevant papers [1, 2], but it should probably be more clear that their ASV idea exists under a different name. - For the claim that ASVs/quasivalues uniquely satisfy Axioms 1-3, I would think that a proof should either be provided or cited, because this result is non-trivial. About the global Shapley values: - These global Shapley values seem to have been introduced by another paper, but the authors' explanation that they quantify contributions to the model's accuracy has a small problem. The probability of the correct class f_y(x) does not quite represent the model's accuracy, at least in the (conventional) sense of 0-1 accuracy. I found the authors' explanation a bit confusing, so they might distinguish more clearly between their view of accuracy and 0-1 accuracy. - The section about applications of these global Shapley values to feature selection raises some questions. ASVs provide a very inefficient tool for checking whether a set of features U has strong predictive power. The authors' explanation suggests that users should 1) calculate global ASVs using the ordering given by Eq. 15 (which is specific to a given U/V partitioning), and then 2) take the sum of the ASVs in U (Eq. 16) to estimate U's predictive power. But because ASVs are not cheap to calculate, obtaining the global ASVs (i.e., average ASVs across *every example in the dataset*) would quite possibly be more computationally costly than training a *single* model on U, which gives you exactly the performance with U. Unless I'm missing something, this seems like a very inefficient approach to feature selection. A more justified approach for applying global Shapley values to feature selection, which does not suffer from this problem, is given by [3]. - Performing feature selection on structured data, like EEG time series, is an odd choice of task. It permits a convenient telescoping of ASVs (Eq. 19), but as a feature selection application this does not make much sense. What purpose could there be for excluding the last n% of an EEG sample, given that the seizure could occur at any point in the time window? - If the authors really want to show that ASVs are useful for feature selection, they might have included an experiment with non-structured data, where feature selection makes more sense. They might also consider comparing to more baselines methods. About the other experiments: - The application of ASVs to the admissions problem does something odd: it assumes that department choice precedes gender, as if department choice is a causal ancestor of gender. That seems inconsistent, because in the previous example (income prediction), attributes like gender were assumed to be causal of all other features. So the causal order is in a sense reversed in this second experiment. - The authors provide some explanation for this choice, saying this use of the ASV framework is intended to distinguish between "resolving variables" (R) and "sensitive attributes" (S). But it's a bit odd to explain this reversal of the causal order in midst of an experiment, since the point of the paper is that ASVs integrate causal knowledge. - Would defining the causal order in a way that is consistent with the previous experiment change the results? - Figure 2 shows that the majority of credit is given by ASVs to the beginning of EEG samples. But calculating global Shapley values for a structured data type is an odd choice, because evidence for a seizure could occur at any point in the sample. These results (Figure 2b in particular) do not provide strong evidence, in my view, that ASVs are the right choice for time series data. It would have been more useful to examine individual samples and show how SV/ASV attributions relate to the regions where individual seizures occur. [1] Monderer and Samet, "Variations on the Shapley value" [2] Weber, "Probabilistic values for games" [3] Covert, Lundberg and Lee, "Understanding global feature contributions through additive importance measures"

Correctness: The claims all seem correct.

Clarity: The paper is very clearly written.

Relation to Prior Work: The authors might consider being more clear about the fact that ASVs are quasivalues, which have been studied extensively in cooperative game theory.

Reproducibility: Yes

Additional Feedback: Update: Thank you for your response, which addressed most of the concerns raised in my review. I think the writing changes you've described will improve the quality of the paper, but after discussing with the other reviewers I have some lingering concerns. The first concern is that it appears inconsistent to sometimes use the causal ordering and sometimes use the anti-causal ordering (as mentioned in my initial review). Previewing this reversal earlier in the paper is helpful, but will not fix the bigger problem that readers don't know when or why to use either approach. The second concern is that it's unclear how ASVs reflect causal information. I share the intuition that this approach makes sense, and I suspect there's a very good justification for it, but I find the current explanation lacking (particularly lines 152-157). Like the other reviewers, I think that improving the paper in this regard would have a large impact.


Review 2

Summary and Contributions: The paper presents a new surrogate model approach to establishing feature importance. It is based on the game theoretic concept of Shapley values to optimally assign feature importances. The Shapley value of a feature’s importance is its average expected marginal contribution after all possible feature combinations have been considered. The paper identifies several shortcomings, being the most important one the fact the Shapley values ignore causality. Based on this observation, the paper identifies one of its axioms as the source of the problem and proposes a new axiom (asymmetric values) in order to consider causal relationships, even in the case where the full causal structure is not known.

Strengths: Up to my knowledge, this is a novel idea that introduces a very interesting point of view. The paper identifies a clear point (causal relationships are very important when considering explainability) that can be mapped to an axiom of the classical method. The modification of this axiom should allow (if correct) the use of causal information in a very elegant way. From an empirical point of view, the experiments are sound and show that this method can offer better insights than the classical one. The method is very relevant to the NeurIPS community and in the cases where we have causal knowledge about datasets, it can become an standard.

Weaknesses: From a theoretical point of view, it is not very clear for the non-specialized reader if the modification of the fourth axiom is relevant with respect to the consistency of the method. A discussion about this point would increase the value of the paper and could contribute to its acceptance by the community. From a more practical point of view, the paper does not discuss what is the impact, if any, of their proposal with regard to the practical computation of the values. The paper does not report about the specific (approximation?) method it is used in the experiments or the complexity of applying the method to high dimensional data. Again, some comments along these lines would increase the confidence for the reader.

Correctness: I have not found anything (theory and methodology) incorrect with the paper. My only suggestion is about adding some analysis about the consistency of the method.

Clarity: The paper is well written, with a clear exposition and good motivation. An interested reader can easily follow the arguments. Limitations of the method are clearly identified.

Relation to Prior Work: The paper proposed a new idea and from this point of view it differs from previous contributions, but like any other paper it is built in previous approaches. These previous approaches are properly recognized and their strengths and weaknesses are fairly discussed.

Reproducibility: Yes

Additional Feedback: Regarding author rebuttal, I appreciate the discussion about the "causality of the prediction process vs causality of data", a topic that must be clarified in the paper, but there are still some weak points (we did not get any explanation about the theoretical consistency of the method). I still believe that this paper deserves to be accepted and remaining open questions can be addressed in the future.


Review 3

Summary and Contributions: Paper proposes asymmetric Shapley values as a way to incorporate causality into explainability.

Strengths: Conceptually a very interesting idea, relatively easy to implement. First attempt to incorporate causality into Shapley values (ignoring [19]).

Weaknesses: Not completely clear from the paper why or when asymmetric Shapley values incorporate causality: this appears to be mainly based on intuition, rather than theoretical argumentation and mathematical principles (despite the claim in the paper, with apologies). The asymmetric approach appears to be in contradiction with [19], which argues that the default Shapley values properly account for causality and no conditioning/on-manifold Shapley values are required. It would be good to discuss where this apparent discrepancy comes from.

Correctness: The argument against symmetry in section 3.1 appears to be a bit circular and then rather weak. The fairness example is rather confusing and hard to generalize. Here the ordering appears to be taken against instead of in the causal direction. The global ASVs then pick up that in the "fair" case any relation between gender and admission is fully mediated by other observed variables, whereas in the "unfair" case there is also a direct effect (as far as the model can tell, since X4 is unobserved). The generalization then appears to define "fairness" as there being no direct (unmediated by other observed variables) relationship between the sensitive variables and the outcome. This "hidden" definition of fairness seems rather unsatisfactory. A bit along the same lines, in 4.3 and 4.4 it is clear that the asymmetric Shapley values shift attribution from the features later in the ordering to those earlier in the ordering: by construction the attribution of the features later in the ordering is only considered conditioned upon the features earlier in the ordering. This leads to a different explanation, but it remains unclear why this is the better explanation.

Clarity: Overall, the paper is a pleasure to read. Probably because of the page limit, some relevant information is lacking, e.g., where the (tiny) errorbars in Figure 1 come from.

Relation to Prior Work: By itself, asymmetric Shapley values are not novel. Their application to explainability of machine learning models is. The most relevant papers are cited, but how they are described is suboptimal, e.g.,: [1] also proposes non-empirical on-manifold distributions, not just empirical; [19] discusses how to handle causality in the context of Shapley values, but reaches a completely different conclusion as this paper.

Reproducibility: Yes

Additional Feedback: See above. The idea itself does make a lot of sense, but a more convincing demonstration/explanation of why ASVs incorporate causality or fairness and a more thorough discussion of related approaches could strengthen the paper tremendously. Update after rebuttal: With apologies, the rebuttal did not really change my opinion. When added, the comparison with [19] is perfectly fine (the lack of mentioning concerned me, not the difference itself). My other concerns sadly still stand: I really like the intuitive idea of considering asymmetric Shapley values for model explanation, but do not feel that the authors provide a theoretical explanation of why/how asymmetric Shapley values incorporate causality. The rebuttal does not address this, just says that three of the four properties of Shapley values are still satisfied. In a similar vein, the discussion of fairness lacks rigor, which makes it hard to understand what's really going on. E.g., why are anti-causal asymmetric Shapley values different/better than "just" applying causal reasoning as in [20] and other papers on e.g., counterfactual fairness? And how can the example be generalized into a generic procedure for checking fairness? For the time-series example, I can understand why asymmetric Shapley values shift attribution to the earlier time points, but not why this is the better explanation (which is not commented on in the rebuttal).


Review 4

Summary and Contributions: The paper extends the Shaply value method that is commonly used for explaining machine learning models. Shapley value is an importance score of a given feature value in a prediction. At a high-level, it can be thought as the marginal change in the prediction when the feature is considered by the model. A key property of Shaply value is symmetry: it gives equal scores to feature values that give the same information. This paper argues that this property is undesirable as it fails to consider partial orders between features such as causal relationship. In the case where A causes B, even though A and B give the same information, one may want to put more importance on A. The paper proposes asymmetric Shapley values. And the key idea is to remove combinations/orders of features that violate this order when computing the Shapley values. The paper demonstrated its effectiveness on four applications.

Strengths: 1. Novel idea to incorporate causal information in a popular interpretability method. Both causality and interpretability are well in the scope of NeurIPS and can trigger interesting discussions. 2. Rich applications demonstrating the usefulness of the approach.

Weaknesses: 1. My biggest concern is whether this proposed approach does explain what a model has learned better than the standard Shapley value method. More concretely, in the case where feature A causes feature B, it could be well the case that the model fails to capture this causal relationship and considers these two features equally. If one applies the asymmetric Shapley value method, it will assign a higher score to A. If this is the case, does the new method give the wrong explanation? However, I can see this method to be useful if one's goal is to adjust the input to get a different result rather than understand the model. 2. Related to the above point, the paper shows a variety of interesting applications but it is hard to evaluate how it advances the state-of-the-art. For example, for "causality-based model explanations", how can we know the scores given by the proposed approach explain the model (rather than the data) better than the existing approaches? As another example, can the existing approaches distinguish the unfair model from the fair model in 4.2?

Correctness: In terms of math and the evaluation setup, I don't see any major mistake. For the fairness experiment, I would like to see how the existing approaches perform.

Clarity: Yes. And if more space is allowed, I encourage the authors to add small examples to explain the concepts.

Relation to Prior Work: Yes.

Reproducibility: Yes

Additional Feedback: 1. Please address points in "Weaknesses". 2. Given a causal graph, what is the general guideline in incorporating it in calculating asymmetric Shapley values? From Section 4.2, it seems like simply taking the partial order indicated by the graph doesn't suffice for some cases. 3. Are asymmetric Shapley values harder to compute than the standard Shapley values in practice? After rebuttal: Thanks for the response. I am happy with the response on what the approach actually explains. However, It is still unclear why the approach gives a better explaination than the non-causal approaches. Also, as other reviews have pointed out, the paper doesn't explain clearly why the approach incorporates causal knowledge. I encourage the authors to explore more in these directions.


Review 5

Summary and Contributions: This paper proposes a generalization of the Shapley value (SV). As is well-known, the original definition of SV is based on all possible combinations of the input variables. The authors look at the definition from a different angle. They point out that the entire combination implies the whole symmetry among the input variables and thus cannot handle causal structures dictated by conditional distributions. The proposed asymmetric SV is a restricted version of SV, where some of the combinations are omitted when violating an assumed causal stricture. It seems the causal structure has to be provided as an assumption. The authors provide a careful discussion on which axioms of SV is relaxed. They argue that the generalization they introduced is a minimal but sufficient modification that can be used in a lot of real-world situations.

Strengths: - Clearly articulates compelling research motivation. - Introduced new idea of asymmetric SV. - Provided a careful mathematical description in light of the SV axioms.

Weaknesses: Use non-standard ML terms such as "coalition". The authors should pay attention to the standard terminology in the ML community (NeurIPS is not only for researchers of the game theory). Mathematical definitions look rather loose.

Correctness: Yes, to the best of my understanding.

Clarity: Yes with some caveats (see Weakness).

Relation to Prior Work: Yes.

Reproducibility: Yes

Additional Feedback: This is a fine piece of work. It has introduced a simple but powerful idea that addresses an fundamental issue of the well-known SV. It is not the kind of papers that boast a performance improvement on a commoditized problem with a big table of experimental comparison. NeurIPS used to be the community where many papers of this kind are presented, which I miss.

[Author Response · NeurIPS 2020]

## Reviewer #1

To clarify the origins of ASVs, we will modify lines 146-7: "In the game theory literature, this axiom was first relaxed by [43], which termed the result 'random-order values'; [30] referred to them as 'quasivalues'." We will add references to line 150: "ASVs uniquely satisfy Axioms 1–3 (q.v. Theorems 12 and 13 in [43], or Theorem 3 in [30])."

To clarify the notion of accuracy in the global Shapley sum rule, we will add: "The accuracy of randomly drawing from $f$'s predicted probability distribution is distinct from the accuracy of predicting the max-probability class."

In response to R1's statement that the seizure (cf. Sec 4.3) could occur at any point in the time series: Each time series represents 1 sec, whereas most seizures last 30–120 sec, so a seizure is occurring (or not) for the entirety of each time series. We will add a sentence in the text to clarify this and hope this makes the application seem less odd.

Regarding R1's concern about the inefficiency of ASVs for feature selection, we propose to reframe Sec 4.4 as demonstrating a property of ASVs rather than a primary application.

Please also see lines 29–32 below in our response to R3.

## Reviewer #3

R3's largest concern is that our paper does not discuss the difference between our approach and [19], which appears to reach a conclusion opposite to ours. To clarify, [19] studies the causality of the *prediction process* rather than the *data-generating process*. In particular, see Fig 2 in [19] which shows the causal process considered there: features ($\tilde{X}$'s) $\rightarrow$ model inputs ($X$'s) $\rightarrow$ model output ($Y$). As [19] does not consider causal structure among the features themselves, their conclusions are not relevant for the goal of our work: to incorporate causal structure present in the data into model explainability. We will make the following addition to the end of Sec 3.2:

"The distribution $w(\pi)$ incorporates the user's knowledge of the data's causal structure into explanations of the model's predictions. Note that this is quite distinct from other work [19], which considers the model's prediction process itself to be a causal process (features $\rightarrow$ model inputs $\rightarrow$ model output) and finds ordinary Shapley values to be sufficient to explain that process. In contrast, ASVs incorporate causal structure present in the data itself."

R3 finds ASVs' incorporation of causality to be mainly based on intuition. We would distinguish between: (i) gaining causal knowledge about the data, and (ii) incorporating it into a model explainability algorithm. ASVs are solely focussed on tackling (ii); domain expertise or causal inference should generally be employed for (i). It is ASV's handling of (ii) that we claim is mathematically principled: one preserves the 3 important Shapley axioms by restricting to permutations of features consistent with causality. We will clarify this in our introduction to ASVs.

R3 is correct that the ASVs of Sec 4.2 place gender and department choice out-of-causal ordering. To measure *unresolved discrimination* with ASVs, the causal structure needs to used differently – namely, in reverse – to detect whether a protected attribute is causally mediated by a resolving variable [20]. To forecast this to the reader, we will modify line 160 (just after ASVs' definition) to read: "Alternatively, anti-causal orderings can also lead to specific insights; e.g. in Sec 4.2 we define ASVs that detect unfair model decisions."

R3 questions the definition of fairness in Sec 4.2. That definition does not allow just *any* indirect dependence on the protected attribute: only dependence on the protected attribute that is mediated by an explicitly specified *resolving variable* (like free department choice) is permitted. This is a common definition considered by [20] and others.

R3 stated that addressing the points above "could strengthen the paper tremendously". With the proposed modifications, we hope R3 will deem our paper worthy of acceptance.

## Reviewer #4

R4 wonders whether ASVs explain the model or the data. The answer (cf. Sec 3.3) lies somewhere in between. As R4 states, "ASVs can be useful if one's goal is to adjust the input to get a different model prediction". However, this goal is not in opposition to "understanding the model" – it cannot be done otherwise. We will note this in the text.

R4 wonders how ASVs advance the state-of-the-art. We claim there is currently no state-of-the-art in causality-based model explainability. See e.g. lines 14–23 in our response to R3 above. For a guideline to incorporate a causal graph into ASVs, see Eq 11. Also see lines 29–32 in our response to R3 above.

## References

[19] Janzing et al, "Feature relevance quantification in explainable AI: a causal problem" (2019).
[20] Kilbertus et al, "Avoiding discrimination through causal reasoning" (2017).
[30] Weber, "Probabilistic values for games" (1988).
[43] Monderer & Samet, "Variations on the Shapley value" (2002).


[Meta-Review · NeurIPS 2020]

The Shapley value based methodology for explaining a model considers features as players whose coalitions result in establishing the prediction. Formally, the impact of a feature is estimated as the difference between the average Shapley value of the coalitions containing this feature, and that of the coalitions not containing the features. The paper describes how to refine the Shapley values based on (possibly incomplete) knowledge about the causal diagram relating the features, by averaging the Shapley value restricted to coalitions compatible with this knowledge. This paper generated a heated discussion. The rebuttal did clarify the distinction between the causality of the prediction process vs causality of data, which was helpful. An unaddressed concern is whether and when the causal or the anti-causal ordering should be used; this ambiguity undermines the clarity of the approach. More in-depth argumentation about the intuition (the restriction to coalitions compatible with causal notions) is required. More elaboration about "resolving variables" and "sensitive attributes" will definitely be appreciated. The AC inclines on the positive side; this paper is very original and timely.